# A distributed residue network permits conformational binding specificity in a conserved family of actin remodelers

**Theresa Hwang[1], Sara S Parker[2], Samantha M Hill[2], Meucci W Ilunga[1], Robert A Grant[1], Ghassan Mouneimne[2], Amy E Keating[1,3]\***

[1]Department of Biology, Massachusetts Institute of Technology, Cambridge, United States; [2]Department of Cellular and Molecular Medicine, University of Arizona Cancer Center, University of Arizona, Tucson, United States; [3]Department of Biological Engineering and Koch Institue for Integrative Cancer Research, Massachusetts Institute of Technology, Cambridge, United States

**Abstract** Metazoan proteomes contain many paralogous proteins that have evolved distinct functions. The Ena/VASP family of actin regulators consists of three members that share an EVH1 interaction domain with a 100% conserved binding site. A proteome-wide screen revealed photoreceptor cilium actin regulator (PCARE) as a high-affinity ligand for ENAH EVH1. Here, we report the surprising observation that PCARE is ~100-fold specific for ENAH over paralogs VASP and EVL and can selectively bind ENAH and inhibit ENAH-dependent adhesion in cells. Specificity arises from a mechanism whereby PCARE stabilizes a conformation of the ENAH EVH1 domain that is inaccessible to family members VASP and EVL. Structure-based modeling rapidly identified seven residues distributed throughout EVL that are sufficient to differentiate binding by ENAH vs. EVL. By exploiting the ENAH-specific conformation, we rationally designed the tightest and most selective ENAH binder to date. Our work uncovers a conformational mechanism of interaction specificity that distinguishes highly similar paralogs and establishes tools for dissecting specific Ena/VASP functions in processes including cancer cell invasion.

**\*For correspondence:**
keating@mit.edu

**Competing interest:** The authors declare that no competing interests exist.

## Editor's evaluation

This manuscript describes interesting follow-up studies on one peptide hit (from a protein called PCARE) coming out of a proteome-wide screen for peptides that can bind to the EVH1 domain of ENAH, one of the three highly homologous Ena/VASP actin regulators. Surprisingly PCARE binds to ENAH selectively over the other two members of Ena/VASP family, EVL and VASP. The authors provide a nice explanation for how this selectivity is achieved and develop a peptide PCARE-Dual that specifically binds ENAH more tightly, setting out the stage for developing potent and selective inhibitors for ENAH.

## Introduction

Metazoan signal transduction networks have evolved a high degree of complexity using adapter proteins that are specialized to make many interactions and/or highly specific interactions (*Rowland et al., 2017*; *Zarrinpar et al., 2003a*). Signaling complexity arises in part from a plethora of interaction domain families such as the SH3, SH2, and PDZ domains. The facile recombination and insertion of modular domains to generate diverse protein architectures have enabled the evolution of new signaling circuits (*Pawson and Nash, 2000*).

Many modular interaction domains bind to short linear motifs (SLiMs), which occur as stretches of 3–10 consecutive amino acids in intrinsically disordered regions of proteins. The SLiM-binding specificity profiles of different paralogous members in a family of domains are often highly overlapping, yet individual members can in some cases engage in highly selective interactions (*Xin et al., 2013*; *Hause et al., 2012*). For example, a SLiM in Pbs2 binds only the SH3 domain of Sho1 out of the 27 SH3 domains in yeast to activate the high-osmolarity stress response pathway (*Zarrinpar et al., 2003b*). Conversely, the actin assembly protein Las17 binds promiscuously to many SH3 domains, including Sho1, to drive actin-related processes such as endocytosis in yeast (*Kelil et al., 2016*; *Robertson et al., 2009*).

The Ena/VASP proteins are a family of actin regulators involved in functions ranging from T-cell activation to axon guidance (*Kwiatkowski et al., 2003*). There are three paralogs in mammals: ENAH (often called Mena), VASP, and EVL. All three proteins contain an N-terminal EVH1 interaction domain responsible for subcellular localization and a C-terminal EVH2 domain that polymerizes actin. The two domains are connected by a linker, predicted to be largely disordered, that contains binding motifs for other proteins. The EVH1 domain binds SLiMs with the consensus motif [FWYL]PX$\Phi$P, where X is any amino acid and $\Phi$ is any hydrophobic residue (*Ball et al., 2000*). This sequence, referred to here as the FP4 motif, binds the EVH1 domain as a polyproline type II (PPII) helix.

Ena/VASP proteins have evolved both overlapping and paralog-specific cellular functions. On the one hand, ENAH, VASP, and EVL can all bind to FP4 motifs in lamellipodin to promote actin assembly at the leading edge (*Krause et al., 2004*; *Hansen and Mullins, 2015*). Single deletions of Ena/VASP paralogs lead to mild phenotypic defects in mice, indicating that the paralogs can functionally compensate for each other, whereas triple mutant mice die after proceeding to late embryogenesis (*Aszódi et al., 1999*; *Lanier et al., 1999*; *Kwiatkowski et al., 2007*). On the other hand, the three paralogs participate in distinct pathways. ENAH, alone, promotes haptotaxis of breast cancer cells through fibronectin gradients

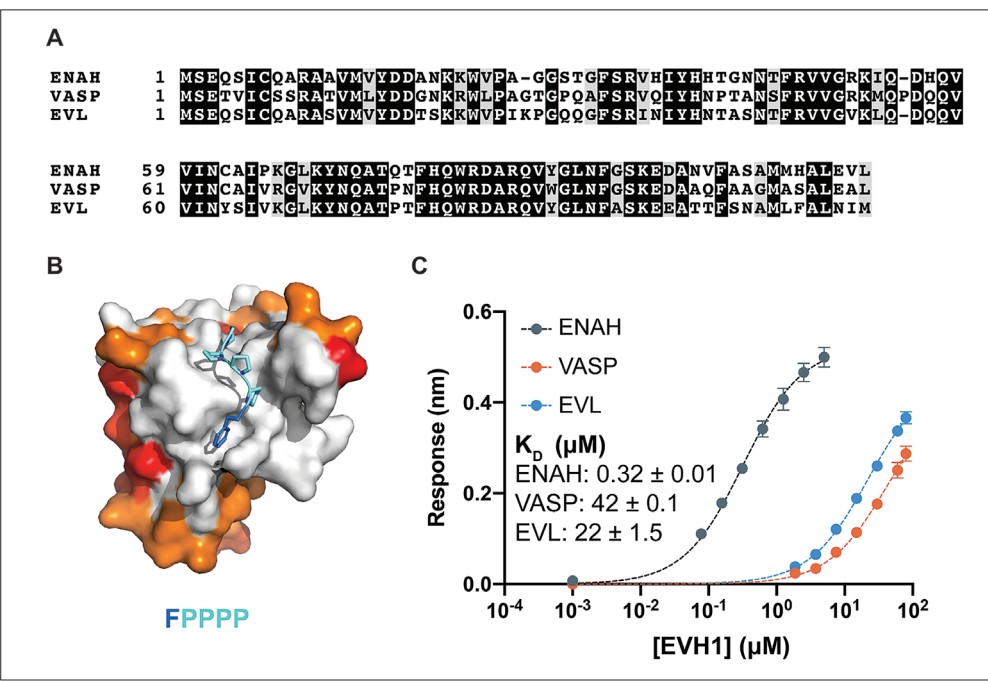

**Figure 1.** Ena/VASP family EVH1 domains are highly conserved but differ in their affinity for a peptide from photoreceptor cilium actin regulator (PCARE). (**A**) Sequence alignment of the EVH1 domains of ENAH, VASP, and EVL. Black denotes residues that are 100 % conserved; residues in gray share similar physicochemical properties. (**B**) Surface representation of ENAH EVH1 bound to an FP4 peptide (PDB 1EVH) highlighting conservation among Ena/VASP paralogs. Residues shared between all three paralogs are white; residues shared by two paralogs are orange, residues that differ in each paralog are red. (**C**) Biolayer interferometry data and curves fit to determine the dissociation constants for peptide PCARE B binding to ENAH, EVL, or VASP EVH1 domains. Error reported as the standard deviation of two replicates.

The online version of this article includes the following figure supplement(s) for figure 1:

**Source data 1.** Raw data for *Figure 1C*.

and regulates translation of specific mRNAs in developing axons, implicating it in functions beyond its role in actin polymerization (*Oudin et al., 2016*; *Vidaki et al., 2017*). In addition, whereas ENAH and VASP promote the invasive potential of migratory breast cancer cells, EVL suppresses breast cancer invasion (*Roussos et al., 2010*; *Zhang et al., 2009*; *Padilla-Rodriguez et al., 2018*).

The FP4-binding pocket of the EVH1 domain is 100 % conserved across ENAH, VASP, and EVL, and the paralogs share 62–72% sequence identity over the entire domain (*Figure 1A, B*). Consequently, the three EVH1 domains recognize many common binding partners. Nevertheless, some proteins bind selectively to certain paralogs. The LIM3 domain of testin binds specifically to the ENAH EVH1 domain in a region adjacent to the highly conserved FP4-binding pocket; this is the only example of an endogenous Ena/VASP EVH1-binding partner where the mechanistic basis for specificity is defined (*Boëda et al., 2007*). Testin binds specifically because it contacts residues that have diverged across paralogs to form distinct surfaces, which is a common mechanism for achieving specificity (*Skerker et al., 2008*; *Bardwell et al., 2009*; *Schreiber and Keating, 2011*).

Using a high-throughput screen of the human peptidome, we identified, among other interaction partners, a peptide from photoreceptor cilium actin regulator (PCARE) that binds with high affinity to the ENAH EVH1 domain. PCARE is almost exclusively expressed in the retina, and mutations in PCARE are associated with inherited retinitis pigmentosa and retinal dystrophy (*Collin et al., 2010*; *Nishimura et al., 2010*; *Kevany et al., 2015*). Prior work implicates PCARE as a scaffold protein important for recruiting complexes that regulate actin assembly as part of new outer segment disk formation in photoreceptor cells (*Corral-Serrano et al., 2020*). Previous investigations identified ENAH as an interaction partner for PCARE based on tandem-affinity purification experiments in HEK293T cells and colocalization of PCARE with Ena/VASP at ciliary membrane expansions in hTERT RPE1 cells (*Boldt et al., 2016*; *Corral-Serrano et al., 2020*). PCARE was also associated, via protein tandem-affinity purification studies, with 17 other proteins with actin-related functions, including Wiskott–Aldrich syndrome protein family member 3 (WASF3), which is recruited via PCARE to the cilium, leading to membrane expansions that involve actin polymerization (*Corral-Serrano et al., 2020*).

Here, we report that PCARE is surprisingly selective for binding to ENAH in preference to EVL or VASP, despite containing an FP4 motif that can engage the perfectly conserved FP4-binding site. We demonstrate that an extended peptide from PCARE can be used as a tool to perturb ENAH-specific interactions in cells. We also reveal the mechanistic basis behind the striking paralog selectivity, which involves an epistatic residue network in the ENAH EVH1 domain that allows ENAH to adopt a conformation that is not accessible to paralogs VASP and EVL. An alpha-helical extension C-terminal to the FP4 motif in PCARE engages and stabilizes this ENAH-specific conformation using a noncanonical binding mode. These observations reveal a strategy to obtain binding specificity in a highly conserved family that must also make promiscuous interactions. Finally, inspired by previous identification of a noncanonical binding site on the backside of the EVH1 domain of VASP (*Acevedo et al., 2017*), which we found is also conserved in ENAH (*Hwang et al., 2021*), we demonstrate how PCARE can be elaborated to give designed synthetic peptides with further enhanced and unprecedented affinity and specificity for ENAH.

## Results

### An extended SLiM from ciliary protein PCARE binds selectively to ENAH EVH1 and reduces adhesion in mammalian cells

In a separate study, we performed a proteomic screen that identified peptides that bind to the ENAH EVH1 domain with dissociation constants primarily in the low- to midmicromolar range ($K_D$ = 2–60 µM). The highest affinity hit from our screen was a 36-residue peptide from PCARE (PCARE[813–848]) that bound to ENAH with a $K_D$ of 0.19 µM (*Hwang et al., 2021*). Truncation studies showed that 23-residue PCARE[826–848], which we call PCARE B, maintains high affinity for ENAH EVH1 ($K_D$ = 0.32 µM). To explore the paralog specificity of PCARE B, we quantified binding to EVL and VASP EVH1 domains using biolayer interferometry (BLI) and discovered a 70- to 140-fold preference for binding to ENAH over EVL or VASP (*Figure 1C*). This was initially very surprising, given that PCARE B includes the FP4 sequence LPPPP and the Ena/VASP paralogs are 100 % conserved in the FP4-binding groove (*Figure 1B*). However, this was also an exciting observation, because it suggested that PCARE B could serve as a reagent for selectively targeting ENAH in cell biological studies of its functions.

To test for association of Ena/VASP proteins and PCARE B in cells, we used Ena/VASP-family-deficient cell line MV$^{D7}$ (**Bear et al., 2000**) and expressed the individual paralogs ENAH, EVL, and VASP. MV$^{D7}$ are embryonic fibroblasts derived from *ENAH$^{-/-}$ VASP$^{-/-}$* mice, and they have low *EVL* expression (**Damiano-Guercio et al., 2020**). We used an shRNA against *EVL* to further decrease residual EVL (**Figure 2—figure supplement 1**). We tagged PCARE B with mRuby2 (mRuby2-PCARE B) and coexpressed this construct with green fluorescent protein (GFP), or GFP fusions to ENAH, an EVH1 deletion mutant of ENAH (ΔEVH1-ENAH), EVL, or VASP. ENAH, EVL, and VASP all robustly localized to focal adhesions as previously observed (**Puleo et al., 2019**) while ΔEVH1-ENAH localization was cytoplasmic. mRuby2-PCARE B exhibited diffuse cytoplasmic localization under all conditions except when coexpressed with ENAH, in which case mRuby2-PCARE B was moderately enriched at focal adhesions, consistent with an ENAH–PCARE B interaction (**Figure 2A**). To demonstrate the specificity of PCARE B for ENAH, we quantified the enrichment ratio of the mean intensity of mRuby2-PCARE B at focal adhesions and in the cytoplasm in each condition. The mean intensity values at focal adhesions and the cytoplasm were divided by the average background intensity of mRuby2-PCARE B. We observed significant enrichment (p < 0.0001) of mRuby2-PCARE B with GFP-ENAH at focal adhesions compared to the cytoplasm. The enrichment ratios for mRuby2-PCARE B with GFP-EVL and GFP-VASP were not significant (**Figure 2B**). In addition, plots of fluorescence intensity along a line passing through focal adhesions showed colocalization of PCARE B and ENAH (**Figure 2C**).

Historically, a mito-tagged ActA peptide has served as a valuable sideways knockout tool that can deplete Ena/VASP proteins from other cytoplasmic locations by recruiting them to the mitochondria. This strategy has been used to dissect Ena/VASP-dependent functions (**Bear et al., 2000**). To test whether PCARE B could serve a similar function, but with specificity for ENAH, we tagged PCARE B with a mitochondrial localization sequence, generating Mito-mRuby2-PCARE B. In MV$^{D7}_{shEVL}$ cells with exogenously expressed Ena/VASP paralogs, ENAH was significantly colocalized with Mito-mRuby2-PCARE B at mitochondria, although some ENAH remained associated with focal adhesions. In contrast, EVL and VASP remained entirely localized to focal adhesions and did not localize to mitochondria (**Figure 2D–F**). There is significant enrichment (p < 0.0001) of Mito-mRuby2-PCARE B with GFP-ENAH at the mitochondria compared to the cytoplasm, whereas the enrichment ratios of Mito-mRuby2-PCARE B with GFP-EVL and GFP-VASP were not significant (**Figure 2E**). Together, these findings indicate that PCARE B can recruit ENAH, but not EVL or VASP, to artificial localization sites such as the mitochondria, supporting interaction specificity between PCARE B and ENAH in cells.

We also tested whether cytoplasmic expression of PCARE B could disrupt the function of endogenous ENAH. We examined focal adhesion maturation as a readout of ENAH function in MCF7 breast cancer cells, which express significant levels of ENAH (**Figure 2—figure supplement 2**). We compared the effects of PCARE B expression to the effects of ENAH knockdown by shRNA by assessing cell adhesion, using paxillin immunofluorescence labeling to delineate focal adhesions (**Figure 2G**). As expected, ENAH knockdown was associated with diminished adhesion and smaller focal adhesion size as compared to cells expressing nontargeting shRNA (**Figure 2F–H**). Intriguingly, expression of PCARE B resulted in a similar decrease in adhesion, suggesting suppression of ENAH function at focal adhesions (**Figure 2G–I**). Importantly, the expression of PCARE B in MCF7 cells shifted the enrichment of ENAH from focal adhesions to membrane protrusions (**Figure 2G** center inset). ENAH enrichment at protrusions was not observed in MCF7 cells expressing Mito-mRuby2-PCARE B, which exhibited strong mitochondrial localization of ENAH and markedly decreased adhesion (**Figure 2G** right inset, **Figure 2H,I**). This suggests that in MCF7 cells, blockade of the EVH1 domain by cytosolic PCARE B liberates ENAH from focal adhesions while permitting other EVH1-independent interactions elsewhere in the cell. In contrast, Mito-mRuby2-PCARE B can recruit ENAH away from its normal sites of action at focal adhesions and the cell membrane and to the mitochondria.

## FP4 motif-flanking elements in ciliary protein PCARE confer high affinity by inducing noncanonical binding

To understand the structural basis for the high-affinity and selective interaction of PCARE B with ENAH, we solved a crystal structure of ENAH EVH1 domain fused to a 36-mer PCARE sequence to 1.65 Å resolution. Fusing a short peptide to its binding domain can assist with structure determination, in part by controlling the stoichiometry of domains and peptides in the crystallization mixture (**Appleton et al., 2006**; **Teyra et al., 2017**; **Li et al., 2019**). To minimize the influence of the fusion

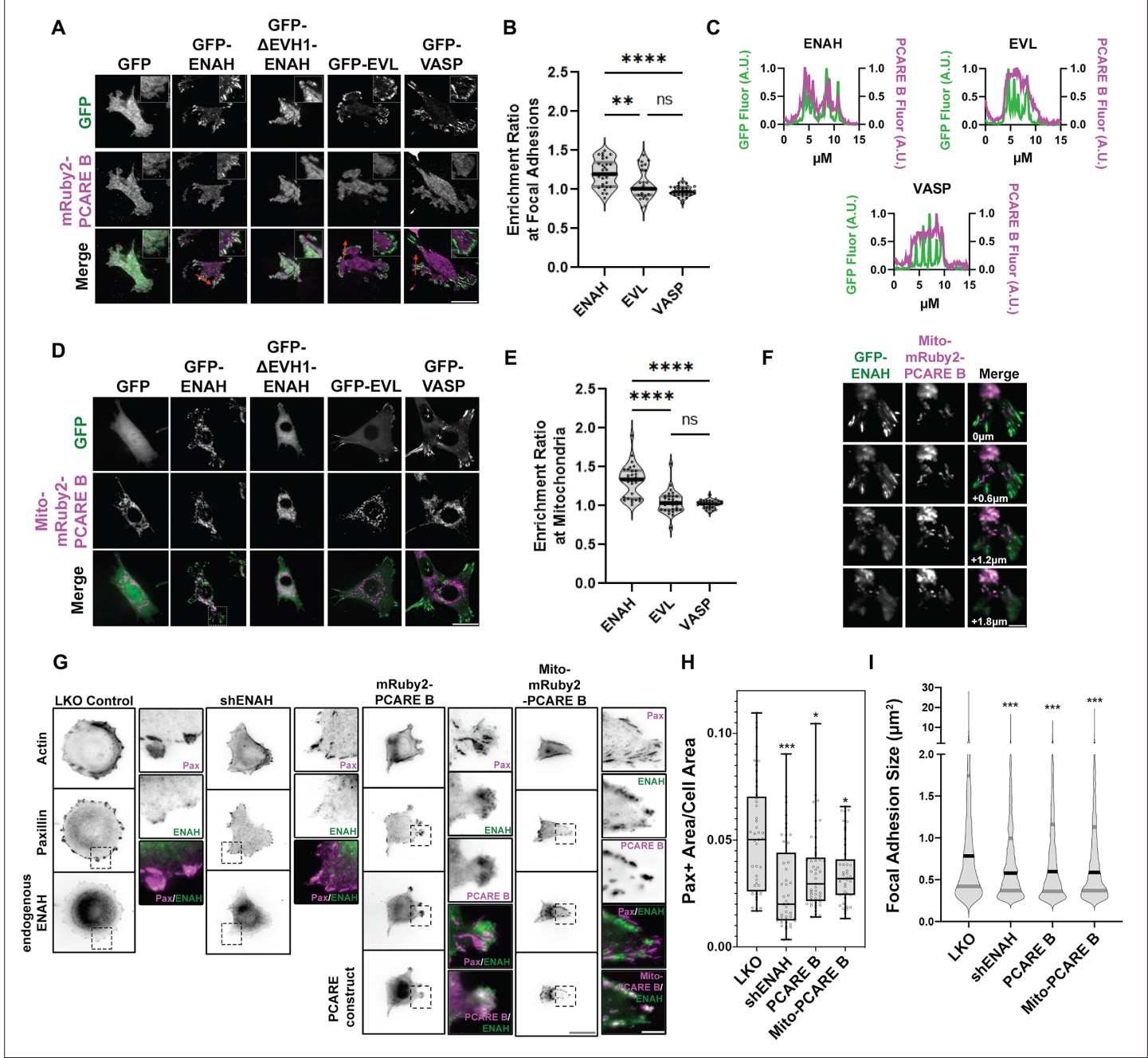

**Figure 2.** A photoreceptor cilium actin regulator (PCARE)-derived peptide selectively recruits ENAH in cells. (**A**) Live MV$^{D7}_{shEVL}$ cells expressing mRuby2-PCARE B with GFP, GFP-ENAH, GFP-dEVH1-ENAH, GFP-EVL, or GFP-VASP, imaged using total internal reflection fluorescence (TIRF) microscopy. Scale bar = 25 µm. (**B**) Enrichment ratio of mRuby2-PCARE B at GFP-positive focal adhesions over cytosolic signal under indicated overexpression conditions. n = 30 focal adhesions, N = 3 biological replicates. (**C**) Normalized fluorescence intensity of GFP signal (left axis) and mRuby2-PCARE B signal (right axis) along a line drawn through focal adhesions, indicated by the red arrow in (**A**). (**D**) Live MV$^{D7}_{shEVL}$ cells expressing Mito-mRuby2-PCARE B with GFP, GFP-ENAH, GFP-dEVH1-ENAH, GFP-EVL, or GFP-VASP. Image is a maximum intensity projection of z-stack acquired using widefield fluorescence microscopy. Scale bar = 25 µm. (**F**) Magnified region of interest indicated by the box in (**D**), showing colocalization of Mito-mRuby2-PCARE B and GFP-ENAH at increasing depths in the cell. Scale bar = 5 µm. (**G**) Immunofluorescence labeling of MCF7 cells expressing nontargeting LKO control, *ENAH*-targeting shRNA, mRuby2-PCARE B, or Mito-mRuby2-PCARE B. Cells were fixed 8 hr after plating and immunolabeled for focal adhesion marker paxillin and endogenous ENAH, and additionally stained with phalloidin for F-actin. Box indicates positions of magnified regions of interest (ROI). Scale bar = 25 µm, magnified ROI scale bar = 5 µm. (**H**) Box-and-whisker plot of total paxillin-positive area per cell normalized to the total cell area, for indicated conditions. N = 3 biological replicates, n = 45–49 cells. (**I**) Violin plot of individual focal

*Figure 2 continued on next page*

*Figure 2 continued*

adhesion size (for adhesions greater than 0.25 µm²) for indicated conditions. The central black line indicates the median, peripheral gray lines indicate interquartile ranges. $N$ = 3 biological replicates, $n$ = 1452–2159 individual adhesions. In panels B, E, H, and I, * $p \leq 0.05$, ** $p \leq 0.01$, *** $p \leq 0.001$, **** $p \leq 0.0001$, Kruskal–Wallis test.

The online version of this article includes the following figure supplement(s) for figure 2:

**Source data 1.** Raw data for *Figure 2B, C, E, H1*.

**Figure supplement 1.** Knockdown of *EVL* in MV[D7] cells.

**Figure supplement 2.** Expression of Ena/VASP proteins in MCF7 cells.

on the domain–peptide interaction, we introduced a long, flexible linker between C-terminus of the ENAH EVH1 domain and the PCARE B sequence. The linker consisted of the 6-residue sequence GGSGSG and then 13 residues of PCARE that lie N-terminal to PCARE B and that we determined using truncation studies are dispensable for high-affinity binding (*Hwang et al., 2021*). Consistent with our truncation studies, our structure showed that only 21 residues of PCARE were fully resolved in the electron density (PCARE[828–848], *Figure 3A*). Notably, neither the Gly and Ser residues nor any of PCARE residues 813–827 gave clear density in our maps, consistent with this part of the fusion protein being disordered and not forming specific interactions that might influence the peptide-binding geometry.

The high-resolution structure led to the surprising discovery that the LPPPP motif in PCARE binds at the expected canonical site, but in the opposite orientation from previously observed Ena/VASP EVH1 domains engaged with proline-rich peptides (*Ball et al., 2000*; *Prehoda et al., 1999*; *Fedorov et al., 1999*; *Figure 3A, B*). PCARE[828–848] uses a 14-residue alpha helix-rich extension C-terminal to the LPPPP motif to make additional contacts to an extended region on the EVH1 domain, explaining its high affinity.

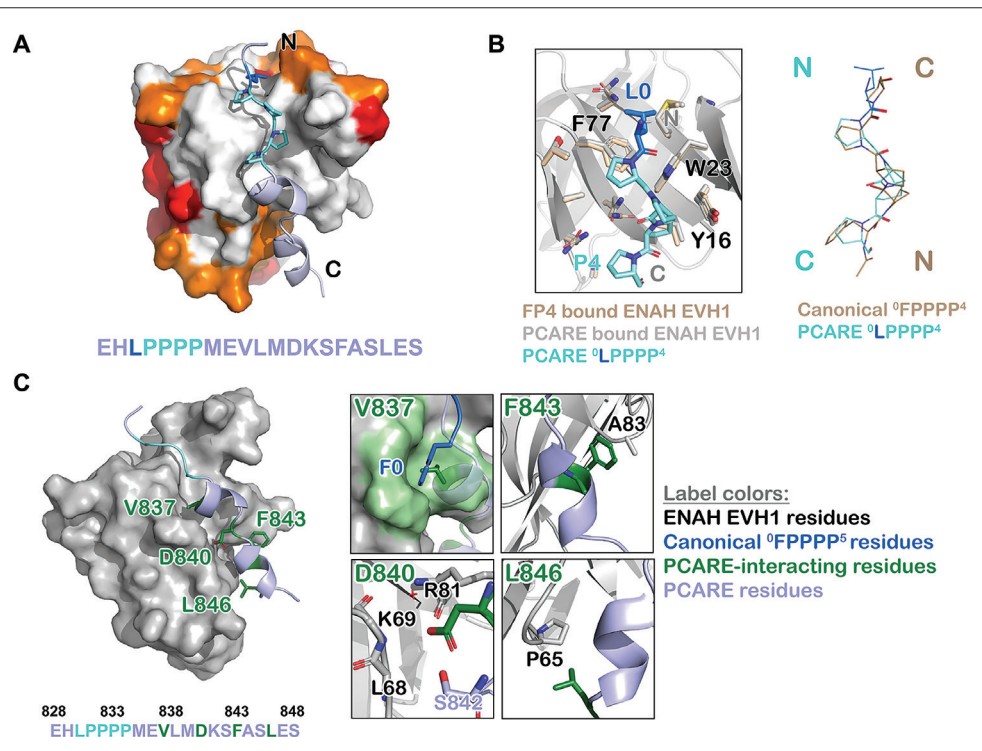

**Figure 3.** ENAH EVH1 binds to a peptide from photoreceptor cilium actin regulator (PCARE) differently than to the peptide FPPPP. (**A**) Surface representation of ENAH EVH1 bound to a segment of PCARE, highlighting Ena/VASP EVH1 domain conservation using colors as in *Figure 1*. (**B**) View comparing the orientations of an FP₄ peptide and the LP₄ region of PCARE. Side chains of the ENAH EVH1 domain are shown as sticks using tan for the FP₄ complex and gray for the PCARE complex. (**C**) Surface representation of ENAH EVH1 domain bound to the PCARE peptide. The LP₄ residues are light blue and other EVH1-interacting residues are green; insets show details of the interactions. Note that the PCARE[828–848] peptide is numbered as 133–153 in the PDB file.

**Table 1.** Affinity of EVL and ENAH EVH1 domain mutants for peptides ActA and PCARE B.

| | ActA $K_D$ (µM)[*, †] | PCARE B $K_D$ (µM)[*] |
|---|---|---|
| ENAH | 5.2 ± 0.2 | 0.32 ± 0.01 |
| EVL | 2.7 ± 0.3 | 22.3 ± 1.5 |
| EVL V65P | 2.4 ± 0.1 | 112.1 ± 9.6 |
| EVL Y62C | 7.1 ± 0.7 | 19.0 ± 2.8 |
| EVL Y62C V65P | 5.8 ± 1.0 | 2.2 ± 0.1 |
| ENAH P65D | 7.2 ± 0.5 | 32.0 ± 3.2 |

[*]Affinities determined by biolayer interferometry (BLI) as described in the methods and in, **Hwang et al., 2021**.

[†]The ActA peptide has sequence FNAPATSEPSSFE**FPPPP**TEDELEIIRETASSLDS (see methods for the exact construct tested).

The online version of this article includes the following source data for table 1:

**Source data 1.** Raw data for **Table 1**.

Contacts between the extended, alpha-helical region of PCARE and ENAH are shown in **Figure 3C**. PCARE residues Phe843 and Leu846 make hydrophobic interactions with Ala83 and Pro65 on ENAH. The side chain of Asp840 on PCARE docks into a polar pocket on ENAH made up of the backbone atoms of ENAH residues Lys69 and Arg81. Notably, the backbone NH and side-chain hydroxyl group of Ser842 on PCARE form hydrogen bonds with the side chain of Asp840, positioning Asp840 to hydrogen bond with a water molecule that is further coordinated by the backbone NH of ENAH Arg81. Most intriguing is the interaction between Val837 on PCARE and the hydrophobic groove in ENAH, which typically engages large aromatic residues. The alpha-helical structure of PCARE[828–848] buries the smaller Val837 in the same site where phenylalanine can bind.

To test whether high-affinity binding of PCARE B in solution is consistent with the interactions that we observe in the structure, we made ENAH EVH1 domain with Pro65 substituted with aspartate (ENAH P65D). Residue 65, shown in **Figure 3C**, is remote from the canonical FP4-binding site and, as expected, this mutation has little effect on the binding of ActA (**Table 1**). However, this single mutation led to a dramatic 100-fold weakening of the affinity of ENAH EVH1 for free PCARE B peptide. This change in affinity is consistent with the contacts observed in our crystal structure playing a critical role in stabilizing peptide binding, and not consistent with a model in which PCARE binds in the previously observed FP4 docking geometry, for example as observed for ActA peptides, which would orient the C-terminal residues that are essential for affinity in the opposite direction (**Prehoda et al., 1999**; **Fedorov et al., 1999**; **Barone et al., 2020**). Collectively, our results reveal a noncanonical mode of binding where the FP4 motif of PCARE binds to the ENAH EVH1 domain in a reversed N-to-C orientation and makes extra contacts to achieve high affinity.

## PCARE achieves paralog selectivity by stabilizing an ENAH EVH1 domain-specific conformation with a novel FP4-flanking sequence element

Interestingly, our structure of ENAH EVH1 domain bound to PCARE[828–848] shows that 16 of the 18 residues that are within 4 Å of PCARE in ENAH are identical in ENAH ,VASP, and EVL. Thus, whereas the FP4-binding site is 100 % conserved, the extended binding site engaged by PCARE is also highly conserved. Residue 63 is alanine in ENAH and VASP, and the corresponding residue 64 in EVL is serine. Modeling serine at position 63 in the PCARE-bound structure of ENAH shows that the side-chain hydroxyl group can be readily accommodated in a solvent-facing conformation without interfering with PCARE binding. On the other hand, residue 65 is proline in ENAH and the corresponding residue is valine in EVL and VASP. As discussed in the previous section, Pro65 is remote from the canonical FP4-binding groove but makes extensive contacts with PCARE and is largely buried at the domain–peptide interface (**Figure 3C**). We speculated that proline vs. valine might contribute to the ENAH-binding preference of PCARE. To test this, we made EVL with a valine-to-proline mutation, with the expectation that this would increase the binding affinity for PCARE B. Surprisingly, EVL V65P bound to PCARE B fivefold weaker than did wild-type EVL ($K_D$ = 112.1 vs. 22 µM, **Table 1**). In contrast, EVL V65P bound to an FP4-containing ActA peptide, which does not contact Val65 (**Barone et al., 2020**), with the same affinity as wild-type EVL ($K_D$ = 2.4 vs. 2.7 µM, **Table 1**), indicating that the V65P mutation does not lead to global disruption of the domain structure but does influence its interaction with PCARE.

Comparing the structures of ENAH EVH1 domain bound to PCARE[828–848] vs. the peptide FPPPP (PDB 1EVH; **Prehoda et al., 1999**) shows a conformational difference in ENAH: a loop composed of residues 80–86, which forms part of the extended PCARE-binding site, is shifted by 3 Å (**Figure 4A**).

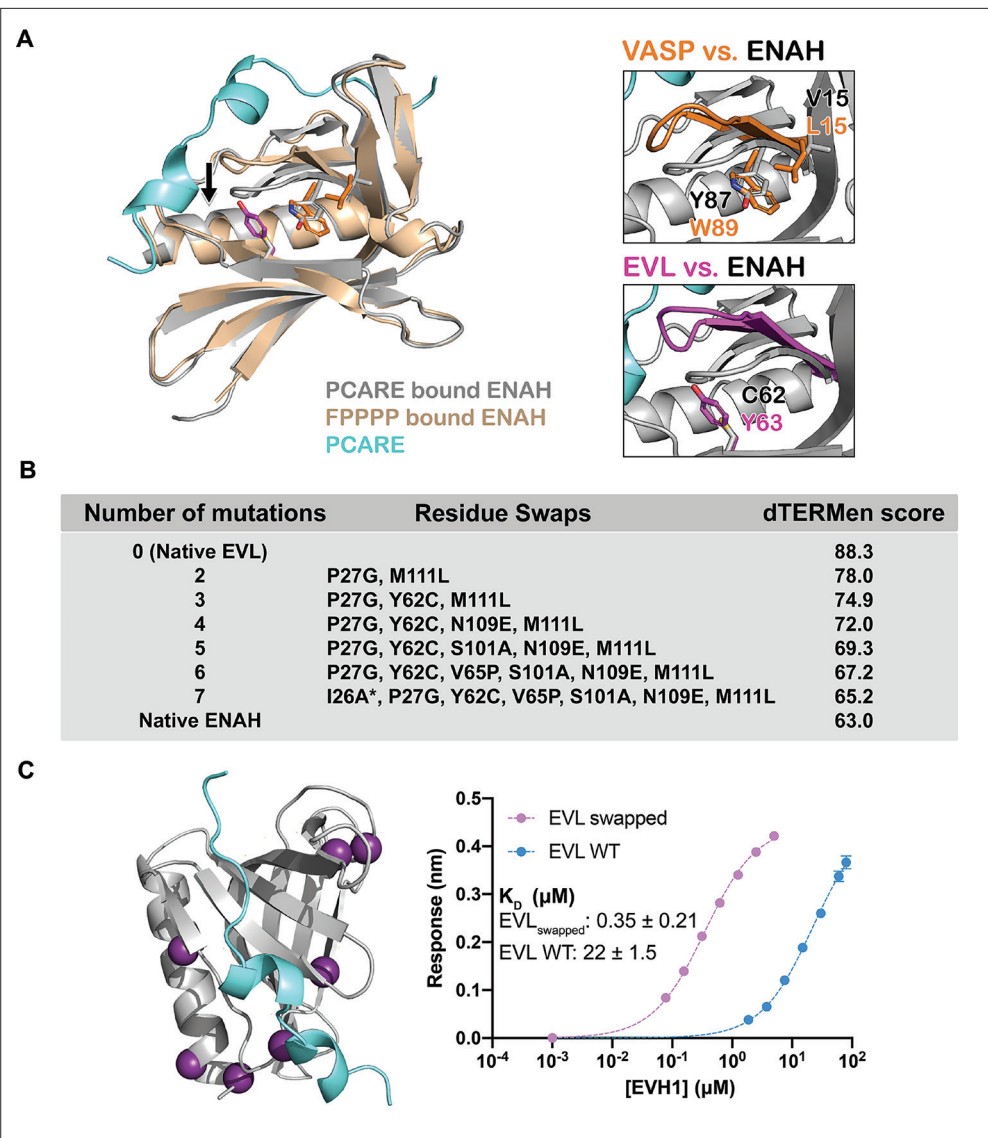

**Figure 4.** A peptide from photoreceptor cilium actin regulator (PCARE) achieves paralog selectivity by stabilizing an ENAH EVH1 domain-specific conformation. (**A**) Superposition of ENAH EVH1 domains bound to PCARE or $FP_4$ peptide (PDB 1EVH). The black arrow highlights a 3 Å shift in a loop that forms part of the binding pocket. Insets show residues that differ between ENAH and VASP or EVL near this loop. (**B**) Lowest dTERMen energy obtained when swapping 0–6 residues from ENAH into EVL, when modeled on the structure of ENAH EVH1 bound to PCARE. * indicates the mutation was added based on manual inspection. (**C**) ENAH EVH1 domain bound to a peptide from PCARE, with residues that were swapped into the EVL EVH1 domain to rescue affinity marked as purple spheres. On the right are binding curves for WT EVL EVH1 domain and EVL_swapped EVH1 domain binding to PCARE B. Error reported as the standard deviation of two replicates.

The online version of this article includes the following figure supplement(s) for figure 4:

**Source data 1.** Raw data for *Figure 4C*.

**Figure supplement 1.** Fast protein liquid chromatography (FPLC) curves for EVL mutants.

Structure gazing suggested that hydrophobic core residues Tyr63 in EVL (Cys62 in ENAH), and Trp89 and Leu15 in VASP (Tyr87 and Val15 in ENAH), are incompatible with this conformational change (*Figure 4A*). To test this, we made EVL EVH1 domain variants EVL Y62C and EVL V65P Y62C. EVL V65P, EVL Y62C, and EVL Y62C V65P EVH1 domains all ran identically as monomers on size-exclusion chromatography (*Figure 4—figure supplement 1*). The Y62C mutation alone had almost no measurable effect on the binding of EVL EVH1 domain to PCARE (*Table 1*), but the EVL Y62C V65P EVH1

domain double mutant bound to PCARE B with $K_D$ = 2.2 µM, which is 10-fold lower than the $K_D$ for wild-type EVL and a striking 56-fold lower than the $K_D$ for binding to EVL V65P (*Table 1*). This dramatic enhancement in affinity for the double mutant relative to either single mutant indicates strong coupling between positions 62 and 65, consistent with the rearrangement of a loop upon PCARE binding. However, these two mutations enhanced binding to EVL EVH1 domain by only 10-fold over wild type, whereas the difference in binding affinity between ENAH and EVL is 70-fold (*Figure 1C*). We concluded that a broader set of residues must contribute to stabilizing the ENAH-specific conformation, but it was not readily apparent which residues these might be.

To identify the ENAH residues responsible for PCARE-binding specificity, we used the structure-based modeling method dTERMen (*Zhou et al., 2020*). dTERMen is a protocol for scoring the compatibility of a sequence with a backbone structure. Energies are computed based on the frequencies with which combinations of residues are found in tertiary motifs in known protein structures. As expected, when scoring different sequences on the structure of ENAH bound to PCARE, the EVL sequence scored considerably worse than the sequence of ENAH itself. Guided by the dTERMen score, we introduced increasing numbers of residues from ENAH into EVL, in an attempt to identify mutations sufficient to confer high affinity binding to PCARE. Seven replacements were sufficient to recapitulate dTERMen energies similar to that for ENAH in the PCARE[828–848]-bound conformation (*Figure 4B*). These residues are distributed across the EVH1 domain, and several are distant from the PCARE[828—848]-binding site (*Figure 4C*). We made a mutated EVL EVH1 domain with the seven corresponding residues from ENAH. This protein, EVL[swapped], bound as tightly to PCARE B as did ENAH EVH1 ($K_D$ = 0.35 vs. 0.32 µM) (*Figure 4C*). Given that wild-type EVL and ENAH differ at 29 sites, and there are

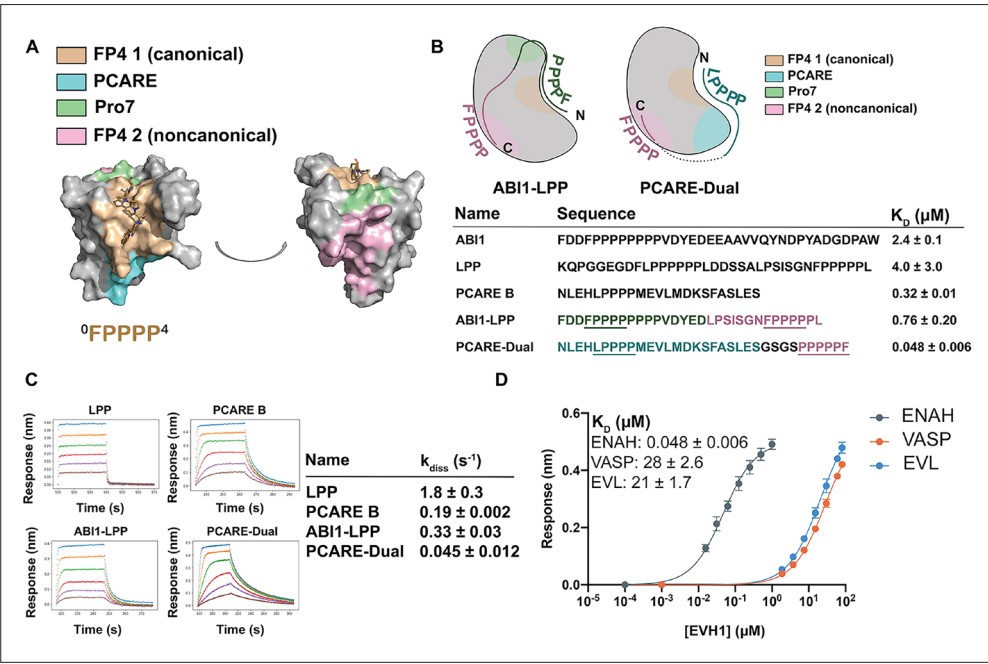

**Figure 5.** Engineered peptides bind to the ENAH EVH1 domain with high affinity and specificity. (**A**) Surface representation of ENAH EVH1 with binding sites discussed in this study indicated. (**B**) Design scheme for high-affinity binders ABI1-LPP and PCARE-Dual. (**C**) Biolayer interferometry (BLI)-binding and dissociation curves. Blue, orange, green, red, purple, and brown curves denote EVH1 concentrations in descending order. LPP: 80, 36, 16, 7.0, 3.1, and 1.4 µM. PCARE B and ABI1-LPP: 2.5, 1.3, 0.63, 0.31, 0.16, and 0.078 µM. PCARE-Dual: 0.50, 0.25, 0.0625, 0.031, 0.016, and 0.0078 µM. Values reported as $k_{diss}$ ± SD for two independent BLI replicates. (**D**) BLI-binding curves for PCARE-Dual binding to the EVH1 domains of ENAH, VASP, or EVL. Errors for (**B**) and (**D**) are reported as the standard deviation of two replicates.

The online version of this article includes the following figure supplement(s) for figure 5:

**Source data 1.** Raw data for *Figure 5B, D*, and *Figure 1*.

**Source data 2.** Raw data for *Figure 5C*.

**Figure supplement 1.** Affinity of PCARE-Dual for ENAH vs. ENAH Y38E.

1.56 million potential residue swaps of 7 residues, it is particularly notable that dTERMen quickly led us, in just a single attempt, to a combination of residues sufficient to transfer binding specificity.

## Engineered binders engage ENAH EVH1 domain with increased affinity and specificity

Based on our structure of PCARE bound to ENAH, we aimed to design even higher affinity, ENAH-selective peptides. To this end, we took a rational design approach. Our strategy relied on designing peptides that can simultaneously engage two binding sites on ENAH EVH1: the canonical FP4-binding site that is occupied by FP4 peptides such as ActA, ABI1, and PCARE (*Prehoda et al., 1999*; *Fedorov et al., 1999*; *Barone et al., 2020*; *Hwang et al., 2021*), and a noncanonical site previously identified in VASP EVH1 that we have shown is also important for certain ENAH–peptide complexes (*Acevedo et al., 2017*; *Hwang et al., 2021*, *Figure 5A, B*). Using a structural model based on PDB structure 5NC7, we estimated appropriate lengths for connecting linkers (*Barone et al., 2020*; *Hwang et al., 2021*). We then made and tested different combinations of binding motifs and linkers that we predicted could bridge these two sites.

In one design, we fused the high-affinity PCARE B sequence, via a short GSGS linker, to a second FP4 motif designed to support bivalent binding while stabilizing the ENAH-specific conformation. Peptide PCARE-Dual bound ~ sevenfold tighter than PCARE B, with $K_D$ = 50 nM (*Figure 5B, D*). Consistent with the designed two-site binding, PCARE-Dual interacted ~ eightfold less tightly with a mutant of ENAH EVH1 domain that had a Y38E substitution in the noncanonical site ($K_D$ = 0.38 ± 0.01 µM; *Figure 5—figure supplement 1*). In a second design, we used a peptide from ABI1 in place of PCARE B and fused this to a peptide from the protein LPP. ABI1 is an Ena/VASP interaction partner that contains the sequence $FP_8$. We have shown that proline residues C-terminal to the FP4 motif, as well as surrounding acidic residues, enhance affinity for the ENAH EVH1 domain (*Hwang et al., 2021*). In particular, the seventh proline of the ABI1 FP4 motif engages ENAH EVH1 at what we call the Pro7 site (green in *Figure 5A, B*). LPP is another ENAH-binding partner that we have shown engages the noncanonical EVH1-binding site. We fused a 17-residue segment of ABI1 to part of the LPP linker and a second FP4 motif to make ABI1-LPP (*Figure 5B*). Our rationale was that the ABI1-derived segment would occupy the canonical FP4-binding site and the LPP linker would wrap along the surface of the EVH1 domain and position a second FP4 motif near the noncanonical FP4 site. The ABI1–LPP fusion peptide bound with $K_D$ = 0.76 µM, which is threefold tighter than the ABI1 portion alone and fivefold tighter than the LPP portion alone, although weaker overall than PCARE-Dual.

The enhancements in affinity for PCARE-Dual and ABI1-LPP come from decreases in off-rate, with PCARE-Dual dissociating ~100- fold slower than dual-motif peptide LPP (*Figure 5C*). Finally, we found that PCARE-Dual is 400- to 600-fold selective for ENAH over EVL and VASP, even though VASP is also known to have a noncanonical binding site that can engage a second FP4 motif (*Acevedo et al., 2017*). To our knowledge, our dual-motif peptides provide the tightest and most specific known binders to the ENAH EVH1 domain to date (*Figure 5D*).

## Discussion

Although the three paralogs have some distinct cellular functions, proteins ENAH, VASP, and EVL are highly conserved in sequence and structure. The EVH1 domains are 100 % identical in sequence in the core FP4-binding groove and share 62–72% sequence identity through the rest of the EVH1 domain (*Figure 1A, B*). FP4-motif peptides engage a small and relatively flat surface on EVH1; previously solved structures have shown how short peptides bind to this core region or, in some cases, to an immediately adjacent hydrophobic patch (*Barone et al., 2020*). The limited contacts with a highly conserved, shallow site make it challenging to achieve high affinity or paralog-selective binding.

There are currently two proteins reported to bind specifically to the ENAH EVH1 domain. The LIM domain of testin binds to ENAH EVH1 with a $K_D$ of 3 µM and does not bind detectably to the EVL or VASP EVH1 domains. Testin achieves paralog specificity by making contacts with ENAH outside of the canonical FP4 groove, at surface sites where the paralogs differ (*Boëda et al., 2007*). Synthetic mini-protein pGolemi binds to ENAH with $K_D$ = 0.29 µM, moderate selectivity (20-fold) over VASP, and higher selectivity (≥120-fold) over EVL (*Golemi-Kotra et al., 2004*). The mechanism behind the specificity of pGolemi has not been determined, and mutational data do not readily rationalize a model (*Holtzman et al., 2007*).

We have now shown that the protein PCARE contains residues adjacent to an FP4 motif that confer both high affinity and selectivity for ENAH over VASP and EVL by stabilizing an ENAH-specific conformation. The biological implications of the specificity of PCARE towards ENAH are unclear at this point. ENAH, VASP, and EVL induce distinct actin remodeling phenotypes based on their differential preferences for profilin isoforms (*Mouneimne et al., 2012*). Thus, paralog selectivity could be necessary when paralog-dependent actin dynamics and structures are required. Both ENAH and VASP have been identified in affinity purification-mass spectrometry experiments using PCARE as bait in ciliated cells (*Corral-Serrano et al., 2020*). Given that ENAH and VASP can heterotetramerize, VASP may be recruited to the identified complexes indirectly, via interaction with ENAH (*Gertler et al., 1996*; *Riquelme et al., 2015*).

A dramatic feature of the PCARE-bound ENAH structure is that the LPPPP motif of PCARE binds in a reversed orientation, relative to previously observed FP4 ligands. The PPII helix possesses twofold rotational pseudosymmetry, meaning that the side chains and backbone carbonyls are similarly positioned in either the N-to-C- or C-to-N-terminal directions. SH3 domains exploit this pseudosymmetry to bind proline-rich sequences in either direction (*Zarrinpar et al., 2003a*). Although all Ena/VASP and Homer EVH1 domain structures solved so far show the PPII helix engaged in a single direction, the WASP EVH1 domain binds its proline-rich ligand in the opposite direction (*Volkman et al., 2002*). Consequently, it has long been hypothesized, although never demonstrated, that Ena/VASP EVH1 domains might also bind FP4 motifs in either direction (*Ball et al., 2002*). Here, we show that this is the case for ENAH. Interestingly, the large effect on PCARE binding that we observed for EVL V65P compared to wild-type EVL EVH1 suggests that EVL also binds PCARE in a reversed orientation (relative to FP4 ligands such as ActA), albeit weakly, because this is the orientation that best explains contacts with position 65.

We are not aware of other examples of such dramatic conformational specificity, in which a natural ligand binds selectively to one paralog despite almost complete (~89%) conservation of the binding site amongst its family members. However, this mechanism is reminiscent of how the cancer drug Gleevec achieves 3000-fold selectivity for Abl over Src, despite the fact that the two proteins share ~46 % sequence identity across the kinase domain, and ~86 % identity in the Gleevec-binding site (*Seeliger et al., 2007*). The mechanism for the selectivity of Gleevec long eluded explanation. As we found here, a few residue swaps based on sequence alignments and structure gazing failed to rescue the affinity of Src for Gleevec, because these mutations failed to account for the higher-order epistatic interactions that contribute to specificity (*Seeliger et al., 2007*). Interestingly, because evolution has sampled kinase sequence space under the constraints of epistasis, retracing the evolutionary trajectories that led to extant Src and Abl kinases, using ancestral reconstruction, allowed Wilson et al. to identify residues involved in a hydrogen-bonding network distant from the Gleevec-binding interface that are key for selective binding to Abl (*Wilson et al., 2015*).

We used an alternative approach to discover a set of residues that contribute to specificity, turning to structure-based modeling. Using dTERMen, we scored the compatibility of different sequences with the ENAH-PCARE[828–848] structure template and identified residues in EVL that are incompatible with this binding mode. Our method solved the challenging problem of identifying mutations within an epistatic network that contribute to function. Importantly, this facile method is easily generalizable to many protein systems as long as a structure is available.

Our results demonstrate an intriguing example of how nature has evolved a protein that achieves selectivity by exploiting a conformation accessible to only one paralog, rather than by making contacts with paralog-specific residues. Our findings raise the question of how widely this mechanism of selectivity is used by other paralogous families, especially within modular interaction domain families. As evidenced by residues identified by dTERMen in our residue swap experiments, hydrophobic core packing plays a key role in enabling new ligand-bound conformations. A recent study demonstrated that randomizing hydrophobic core residues of the SH3 domain from human Fyn tyrosine kinase could effectively switch its affinity and specificity to different ligands (*Ben-David et al., 2019*). In addition, directed evolution experiments that varied only the hydrophobic core residues of ubiquitin yielded a conformationally stabilized ubiquitin variant that was specific for the deubiquitinase USP7 (*Zhang et al., 2013*). These results hint that such mechanisms of conformational specificity could be widespread.

The PCARE peptide identified in this study will serve as a valuable research tool to dissect ENAH-specific biological functions in cells. Cytoplasmic expression of a PCARE peptide displaces ENAH from focal adhesions but not membrane protrusions, indicating that this peptide could serve as a valuable tool

to selectively perturb EVH1-dependent functions of ENAH. In contrast, a mito-tagged PCARE peptide selectively sequesters ENAH at the mitochondria, depleting it from sites of normal function and inhibiting ENAH-dependent biological processes such as cell adhesion. This is likely due to the high avidity of these peptides when localized to the mitochondrial surface, which could potentially outcompete interactions between the EVH2 domain of ENAH and actin-based structures (*Bachmann et al., 1999*).

There is emerging evidence in the literature that paralog-specific functions exist within the Ena/VASP family. For example, ENAH, but not VASP or EVL, is a regulator of local translation in neurons (*Vidaki et al., 2017*). In contrast, EVL has a specific role in durotactic invasion (*Puleo et al., 2019*). Our mito-tagged peptides will serve as valuable research reagents in uncovering and characterizing ENAH-specific molecular functions in other tissues, such as the retina. Interestingly, although VASP is phosphorylated at Tyr39 by c-Abl, we did not find direct evidence in the literature for corresponding phosphorylation of ENAH at the analogous position Tyr38 (*Maruoka et al., 2012*). ENAH is a substrate of c-Abl, but experiments by Tani et al. using transfected murine ENAH or ENAH mutants, c-Abl, and the adapter protein Abi1 showed that the ENAH modification site is Tyr 296; the mutation Y296F ablated the phosphorylation signal, indicating that Y38 is not a site of modification, at least under the tested conditions (*Tani et al., 2003*). This has implications for the mechanism of specificity of our dual-motif inhibitors, such as PCARE-Dual, which we showed binds eightfold weaker to ENAH Y38E (*Figure 5—figure supplement 1*). Cellular conditions in which there is phosphorylation of the backside site of VASP (and perhaps EVL) but not ENAH could further increase the specificity of inhibitor targeting.

The role of ENAH in cancer metastasis has motivated work to identify inhibitors of its EVH1-mediated interactions, and extensive structure-based design and chemical optimization recently led to a high-affinity molecule ($K_D$ = 120 nM) that reduced breast cancer cell extravasation in a zebrafish model. This compound and its analogs mimic the PPII conformation of an FP4 motif. However, because the molecule binds to the highly conserved FP4-binding site, this inhibitor has similar affinity for ENAH, EVL, and VASP (*Barone et al., 2020*). Paralog-specific Ena/VASP inhibitors could have higher therapeutic potential, given the antagonistic roles of ENAH and EVL in promoting and suppressing breast cancer metastasis. ENAH deficiency in mouse models of cancer has been shown to decrease metastasis by reducing tumor cell invasion and intravasation (*Roussos et al., 2010*). On the other hand, EVL is known to promote the formation of cortical actin bundles that suppress the invasion of breast cancer cells (*Padilla-Rodriguez et al., 2018*). Thus, developing high-affinity, paralog-specific inhibitors of Ena/VASP proteins could be crucial in reducing pleiotropic side effects when treating metastasis. Although there are many obstacles to developing peptides as therapeutic inhibitors of cytoplasmic targets, vigorous research on a wide range of strategies for delivering peptides and even complete proteins into cells is ongoing (*Lu et al., 2020*; *Yu et al., 2021*). Independent of these advances, the exquisite affinity and specificity of the PCARE B peptide for ENAH firmly establishes the possibility of paralog-selective targeting in this family, providing key proof of principle for the development of conformationally specific therapies based either on peptides or peptide-mimicking small molecules to treat ENAH-dependent diseases.

## Materials and methods

### Key resources table

| Reagent type (species) or resource | Designation | Source or reference | Identifiers | Additional information |
|---|---|---|---|---|
| Strain, strain background (*Escherichia coli*) | DH5a | NEB | Cat# 2987H | Chemically competent cells |
| Strain, strain background (*Escherichia coli*) | BL21(DE3) | Novagen | Cat# 71400 | Chemically competent cells |
| Cell line (*Homo sapiens*) | MCF7 | Joan Brugge, Harvard | | |
| Cell line (*M. musculus*) | MV^D7 | *Bear et al., 2000* | | |
| Cell line (*Homo sapiens*) | HEK293T | ATCC | | |
| Antibody | anti-ENAH (rabbit polyclonal) | Sigma | Cat# HPA028448 | (1:100) |
| Antibody | anti-Paxillin (mouse monoclonal) | BD Biosciences | Cat# 612405 | (1:200) |

*Continued on next page*

*Continued*

| Reagent type (species) or resource | Designation | Source or reference | Identifiers | Additional information |
|---|---|---|---|---|
| Antibody | Goat anti-mouse Alexa Fluor 488 (goat polyclonal) | Thermo Fisher | | (1:1000) |
| Antibody | Goat anti-mouse Alexa Fluor 647 (goat polyclonal) | Thermo Fisher | | (1:1000) |
| Antibody | Donkey anti-rabbit Alexa Fluor 405 (donkey polyclonal) | Abcam | | (1:1000) |
| Antibody | SureLight Allophycocyanin-anti-FLAG antibody (mouse monoclonal) | Perkin Elmer | Cat# AD0059F | (1:100) |
| Antibody | anti-actin (mouse monoclonal) | ProteinTech Group | | (1:2500) |
| Antibody | anti-GAPDH (mouse monoclonal) | Cell Signaling Technology | Cat# 5174S | (1:000) |
| Antibody | anti-EVL (rabbit polyclonal) | Sigma | Cat# HPA018849 | (1:1000) |
| Antibody | anti-VASP (rabbit monoclonal) | Cell Signaling Technology | Cat# 3132S | (1:1000) |
| Peptide, recombinant protein | Streptavidin, R-Phycoerythrin Conjugate (SAPE) | Thermo Fisher | Cat# S866 | (1:100) |
| Chemical compound, drug | Alexa Fluor 488-Phalloidin | Thermo Fisher | | (1:40) |
| Chemical compound, drug | Alexa Fluor 647-Phalloidin | Thermo Fisher | | (1:40) |
| Recombinant DNA reagent | pLKO.1-Enah shRNA (plasmid) | GE Dharmacon | | |
| Recombinant DNA reagent | pLKO.1-Evl shRNA (plasmid) | Sigma | | |

Protein sequences can be found in *Supplementary file 1*.

## Protein expression and purification

Monomeric ENAH, EVL, or VASP EVH1 domains for use in BLI and ITC experiments, and ENAH EVH1–peptide fusions for crystallography were cloned into a pMCSG7 vector (gift from Frank Gertler, MIT), which places a 6x-His and TEV cleavage tag N-terminal to the EVH1 domain. These constructs were transformed into Rosetta2(DE3) cells and grown in 2xYT media supplemented with 100 µg/ml ampicillin. Cells were grown while shaking at 37 °C to an O.D. 600 of 0.5–0.7 and then cooled on ice for at least 20 min. Cells were then induced with 0.5 mM IPTG and grown while shaking at 18 °C overnight. Induced cultures were resuspended in 25 ml of wash buffer (20 mM 4-(2-hydroxyethyl)-1-piperazineethanesulfonic acid (HEPES ) pH 7.6, 500 mM NaCl, 20 mM imidazole) and frozen at −80 °C overnight. The next day, cultures were sonicated, spun down, and applied to Ni-NTA agarose resin equilibrated with wash buffer, and washed as described above. Samples were eluted in 10 ml elution buffer (20 mM HEPES pH 7.6, 500 mM NaCl, 30 mM imidazole). TEV protease was added to the elution at a ratio of 1 mg TEV:50 mg tagged protein along with 1 mM dithiothreitol(DTT). This mixture was dialyzed against TEV cleavage buffer (50 mM HEPES pH 8.0, 300 mM NaCl, 5 mM DTT, 1 mM EDTA) at 4 °C overnight and then applied over Ni-NTA agarose resin equilibrated with wash buffer. The column was washed with 2 × 8 ml of wash buffer, and the resulting flow-through and washes were pooled, concentrated, and applied to an S75 26/60 column equilibrated in gel filtration buffer (20 mM HEPES pH 7.6, 150 mM NaCl, 1 mM DTT, 1 % glycerol). Purity was verified by sodium dodecyl sulfate-polyacrylamide gel electrophoresis (SDS–PAGE) and combined fractions were concentrated and flash frozen at −80 °C.

## Small-scale protein purification for Biolayer Interferometry

SUMO–peptide fusions were cloned into a pDW363 vector that appends a biotin acceptor peptide (BAP) sequence and 6x-His tag to the N-terminus of the protein and transformed into Rosetta2(DE3) cells (Novagen). Rosetta2(DE3) cells encoding SUMO–peptide fusions were grown in 20 ml of LB +100 µg/ml ampicillin + 0.05 mM D-(+)-biotin dissolved in DMSO for in vivo biotinylation. Cells were grown to an OD of 0.5–0.7 with 1 mM IPTG and induced for 4–6 hr at 37 °C. Pellets were spun

down and frozen at −80 °C for at least 2 hr. Pellets were then thawed and resuspended in B-PER reagent (Thermo Fisher) at 4 ml/g of pellet with 0.2 mM PMSF. This suspension was shaken at 25 °C for 10–15 min and then spun down at 15,000 g for 10 min. The supernatant was applied to 250 µL of Ni-NTA agarose resin equilibrated in 20 mM Tris pH 8.0, 500 mM NaCl, 20 mM imidazole (Buffer A), and then washed three times with 1 ml of this buffer. Peptides were eluted in 1.8 ml of 20 mM Tris pH 8.0, 500 mM NaCl, 300 mM imidazole to use in BLI assays.

## Biolayer Interferometry

All BLI experiments were performed on an Octet Red96 instrument (ForteBio). Biotinylated, 6x-His-SUMO–peptide fusions purified in small scale were diluted into BLI buffer (phosphate-buffered saline [PBS] pH 7.4, 1 % Bovine Serum Albumin (BSA), 0.1 % Tween-20, 1 mM DTT) and immobilized onto streptavidin-coated tips (ForteBio) until loading reached a response level between 0.5 and 0.6 nm. The loaded tips were immersed in a solution of ENAH EVH1 domain at a relevant dilution series in BLI buffer at an orbital shake speed of 1000 rpm and data were collected until the binding signal plateaued. ENAH-bound tips were subsequently placed into BLI buffer for dissociation and data were collected until the binding signal plateaued. $K_D$ values were obtained through steady-state analysis. Briefly, the data were corrected for background by subtracting the signal obtained when doing the same experiment using biotinylated, 6x-His-SUMO lacking any peptide, instead of an immobilized ENAH-binding peptide. The association phases were then fit to a one-phase-binding model in Prism and the equilibrium steady-state-binding signal values from that fit were plotted against ENAH concentration and fit to a single-site-binding model in Prism to obtain dissociation constants. Errors are reported as the standard deviation of two replicates.

## Crystallography

Crystals of ENAH fused at the C-terminus to PCARE were grown in hanging drops containing 0.1 M Tris pH 8.0 and 3.30 M NaCl at 18 °C. 1.5 µl of ENAH-PCARE (769 µM in 20 mM HEPES, 150 mM NaCl, 1 mM DTT) was mixed with 0.5 µl of reservoir solution, and football-shaped crystals appeared in 2 days. Diffraction data were collected at the Advanced Photon Source at Argonne National Laboratory, NE-CAT beamline 24-IDE. The ENAH-PCARE data were integrated and scaled to 1.65 Å with XDS, and the structure was solved with molecular replacement using the ENAH EVH1 structure PDB 6RD2 as a search model (*Barone et al., 2020*). The structure was refined with iterative rounds of model rebuilding with PHENIX and COOT (*Adams et al., 2010*; *Emsley et al., 2010*). *Supplementary file 2* reports refinement statistics. The structure is deposited in the PDB with the identifier 7LXF. Note that the PCARE[828–848] peptide is numbered as 133–153 in the PDB file in accordance with the ENAH–PCARE fusion protein numbering.

## Modeling using dTERMen

The dTERMen scoring function and protocol are described in *Zhou et al., 2020*. The method requires that a template-specific scoring function be computed, based on statistics derived from structures in the PDB. After that, the inputs to dTERMen are the backbone coordinates of a structure and a sequence. dTERMen returns a score for the input sequence adopting the input structure; lower scores correspond to lower energies. Side-chain positions are not modeled explicitly. To score the EVL sequence on the ENAH-PCARE backbone template, we generated pairwise alignments of the EVH1 domains of ENAH and EVL to determine how to map sequence to structure. The EVL EVH1 domain is longer than that of ENAH by one residue, so Lys27 was not included in modelingLys27 was also removed from the EVL V65P, EVL Y62C, and EVL V65P Y62C mutants, that were tested (*Supplementary file 1*).

We then used dTERMen to score all possible combinations of residue swaps between EVL and ENAH, up to six possible positions. Residue swap combinations that led to the minimum energy score were recorded. The best six mutations were sufficient to nearly recapitulate the energy score of the native ENAH sequence on the ENAH-PCARE template. We also included an I26A mutation based on manual inspection of the ENAH-PCARE[828–848] structure, which also lowered the dTERMen energy. We cloned, overexpressed, and purified this swapped EVL sequence, as described in *Hwang et al., 2021*, to test for binding to PCARE B.

## Plasmids for cell culture

For experiments in mammalian cells, the following plasmids were used: pLKO.1-Enah shRNA (GE Dharmacon TRCN0000061827, *Homo sapiens* antisense 5'-TTAGAGGAGTCTCAACAGAGG-3'), pLKO.1-Evl shRNA (Sigma TRCN0000091075, *Mus musculus* antisense 5'- TTGTTCATTTCTTCCA TGAGG-3'), and nontargeting pLKO.1 control (a gift from Felicia Goodrum, University of Arizona). GFP, and mouse cDNA sequences for GFP-tagged ENAH, VASP, and EVL (gifts from Frank Gertler, MIT) were subcloned into the pCIB lentiviral expression vector (Addgene plasmid #120862) as previously described (*Puleo et al., 2019*). ENAH EVH1 domain deletion mutant was generated using inverse PCR site-directed mutagenesis of full-length ENAH. All sequences of constructed plasmids were confirmed by Sanger sequencing. mRuby2-PCARE B and Mito-mRuby2-PCARE B inserts were synthesized (Twist Bioscience) and subcloned into a SFFV-promoter lentiviral expression vector.

## Cell culture

MCF7 and HEK293T cells were cultured in high-glucose Dulbecco's modified Eagle's medium (DMEM) base media with sodium pyruvate (Corning) supplemented with 10 % fetal bovine serum (FBS; Millipore), 2 mM L-glutamine (Corning), and 100 U/ml penicillin with 100 µg/ml streptomycin (Corning). MCF7 cells were confirmed mycoplasma negative and validated by STR testing through the Arizona Cancer Center EMSR core facility. $MV^{D7}$ cells were cultured in high-glucose DMEM supplemented with 15 % FBS, 2 mM L-glutamine, 100 U/ml mouse interferon-γ (Millipore). MCF7 and HEK293T cells were maintained in a 37 °C humidified incubator under 5 % $CO_2$. $MV^{D7}$ cells were maintained in a 32 °C humidified incubator under 5 % $CO_2$. For lentiviral production, second-generation lentiviral particles were generated by PEI transfection of 293T cells as previously described (*Yang et al., 2017*) with transfer plasmid, pMD2.G, and psPAX2 (Addgene #12259, #12260, gifts from Didier Trono). HEK293T media containing lentiviral particles was collected, filtered, and added directly to cultures with polybrene (Gibco). Puromycin (2 µg/ml final concentration; Thermo Fisher) and blasticidin (4 µg/ml final; Gibco) were used to select for cells stably expressing shRNA sequences or Ena/VASP constructs, respectively, after lentiviral transduction. For immunofluorescence experiments, MCF7 cells were cultured on fibronectin-coated coverslips (10 µg/ml; Corning). For live cell imaging ($MV^{D7}$), cells were cultured in fibronectin-coated glass-bottom dishes (Mattek).

## Reverse-transcription quantitative PCR

To assess *EVL* knockdown in $MV^{D7}$, total cellular RNA was isolated using the Isolate II RNA kit (Bioline) according to the manufacturer's instructions. cDNA was synthesized from 1000 ng of input RNA using qScript cDNA Synthesis kit (Quantabio). Reverse-transcription quantitative PCR reactions were run in duplicate on an ABI 7500 Fast Real-Time PCR System (Applied Biosystems) with PowerTrack SYBR Green Master Mix (Thermo Fisher). Primer pairs were confirmed to have 85–110% efficiency based on the slope of the standard curve from a cDNA dilution series. $C_T$s were normalized to the $C_T$ *GAPDH* housekeeping genes. Percent knockdown was determined using the comparative $C_T$ method. *M. musculus GAPDH* Fwd AGGTCGGTGTGAACGGATTTG, Rev GGGGTCGTTGATGGCAACA. *EVL* Fwd TGAGAGCCAAACGGAAGACC, Rev TTCTGGACAGCAACGAGGAC.

## Western blotting

Cells were lysed in buffer containing 140 mM NaCl, 10 mM Tris pH 8.0, 1 mM EDTA, 0.5 mM EGTA, 1 % Triton X-100, 0.1 % sodium deoxycholate, and 0.1 % SDS with protease and phosphatase inhibitors (Boston Bio Products). Samples were resolved by SDS–PAGE and transferred onto nitrocellulose membranes. Membranes were blocked in Odyssey Blocking Buffer (LI-COR) for 1 hr and incubated at 4 °C overnight with primary antibodies. Primary antibodies were used as follows: mouse Actin 1:2,500 (ProteinTech Group, 66009-1-Ig), mouse GAPDH 1:1000 (Cell Signaling Technology, 5174 S), rabbit ENAH 1:250 (Sigma, HPA028696), rabbit EVL 1:1000 (Sigma, HPA018849), rabbit VASP 1:1000 (Cell Signaling Technology, 3132 S). Membranes were incubated with secondary antibodies conjugated to either Alexa Fluor 680 or 790 (Thermo Fisher) for 1 hr. Immunoblots were scanned using Odyssey CLx imager (LI-COR).

## Immunofluorescence

MCF7 cells were fixed and immunolabeled 8 hr after plating onto fibronectin (10 µg/ml; Corning) coated coverslips to assay focal adhesions. Cells were fixed with 4 % paraformaldehyde (PFA; Electron Microscopy Services) with 0.075 mg/ml saponin (*Alfa Aesar*, Sigma) diluted in PBS at 37 °C for 10 min. PFA was quenched with 100 mM glycine in PBS at room temperature for 10 min. Cells were then blocked in 1 % BSA plus 1 % FBS in PBS either overnight at 4 °C or for 1 hr at room temperature. The following immunofluorescence reagents and antibodies were used: mouse anti-Paxillin (1:200, clone: 349; BD Biosciences, 612405), rabbit anti-ENAH (1:100; HPA028448, Sigma), goat anti-mouse Alexa Fluor 488 and Alexa Fluor 647 (1:1000, Thermo Fisher), donkey anti-rabbit Alexa Fluor 405 (Abcam), and Alexa Fluor 488- and Alexa Fluor 647-Phalloidin (1:40, Thermo Fisher). Primary antibodies were diluted in block solution and incubated for 1.5 hr at room temperature. After washing, coverslips were incubated for 1 hr in secondary antibody solution with fluorescently labeled phalloidin. Coverslips were mounted using ProLong Gold Antifade (Invitrogen) and allowed to cure for at least 24 hr before imaging.

## Microscopy and image analysis

Cells were imaged on a Ti-E inverted microscope (Nikon), with a ×100 Apo TIRF 1.49 NA objective (Nikon) and an ORCA-Flash 4.0 V2 CMOS camera (Hamamatsu). For focal adhesion assessment, cells were imaged with total internal reflection fluorescence (TIRF) microscopy. To increase the depth of imaging for examination of mitochondrial localization, standard widefield fluorescence microscopy was used. Focal adhesion quantification was performed as previously described (*Puleo et al., 2019*). Briefly, a binary mask was generated for paxillin signal and actin signal, denoting focal adhesions and cell area, respectively. To facilitate semiautomated segmentation of focal adhesions, we generated a sharp, high contrast image of the paxillin and actin channels by the following processing steps: deconvolution using five iterations of the Richardson–Lucy algorithm, shading correction using rolling ball, and unsharp masked (NIS Elements). Focal adhesion area was quantified by measuring the paxillin area of each cell within the whole cell area, or by examining individual focal adhesions. Enrichment ratio was quantified by measuring the mean fluorescence intensity of mRuby2-PCARE B, Mito-mRuby2-PCARE B, or GFP-ENAH/EVL/VASP at regions of interest and normalizing to average cytosolic intensity. Each experimental condition was performed in triplicate and plotted together. Images presented in figures have been lightly processed in NIS Elements, including by applying 2 iterations of the Richardson–Lucy deconvolution algorithm and rolling ball shading correction to reduce background in live cell images.

## Acknowledgements

This project was supported by NIGMS award R01 GM129007 to AEK and by NCI award R01 CA196885-01 to GM. Part of this work has been supported by Koch Institute NCI Cancer Center (Core) Support Grant (CCSG) P30-CA14051. Part of this work is based upon research conducted at the Northeastern Collaborative Access Team beamlines, which are funded by the National Institute of General Medical Sciences from the National Institutes of Health (P30 GM124165). The Eiger 16 M detector on the 24-ID-E beamline is funded by an NIH-ORIP HEI grant (S10OD021527). This research used resources of the Advanced Photon Source, a U.S. Department of Energy (DOE) Office of Science User Facility operated for the DOE Office of Science by Argonne National Laboratory under contract DE-AC02-06CH11357. TH was partially supported by NIGMS T32 GM007287 and a fellowship from the Koch Institute for Integrative Cancer Research.

We thank the MIT Structural Biology Core for assistance with X-ray crystallography and the MIT Biophysical Instrumentation Facility for instrumentation resources. We thank L Backman for help with X-ray data collection. We thank F Gertler and J Tadros for constructs. We thank members of the Keating lab, Mouneimne lab, and F Gertler for their thoughtful input on the manuscript.

## Additional information

### Funding

| Funder | Grant reference number | Author |
|---|---|---|
| National Institute of General Medical Sciences | GM129007 | Amy E Keating |
| National Cancer Institute | CA196885-01 | Ghassan Mouneimne |
| National Institute of General Medical Sciences | GM007287 | Theresa Hwang |
| Koch Institute for Integrative Cancer Research | Fellowship | Theresa Hwang |

The funders had no role in study design, data collection and interpretation, or the decision to submit the work for publication.

### Author contributions

Theresa Hwang, Conceptualization, Data curation, Formal analysis, Writing – original draft, Writing – review and editing; Sara S Parker, Conceptualization, Data curation, Formal analysis, Resources, Writing – original draft, Writing – review and editing; Samantha M Hill, Data curation, Formal analysis, Writing – review and editing; Meucci W Ilunga, Data curation; Robert A Grant, Data curation, Formal analysis, Supervision, Writing – review and editing; Ghassan Mouneimne, Conceptualization, Funding acquisition, Resources, Supervision, Writing – review and editing; Amy E Keating, Conceptualization, Funding acquisition, Investigation, Project administration, Resources, Supervision, Writing – original draft, Writing – review and editing

### Author ORCIDs

Theresa Hwang http://orcid.org/0000-0002-9011-2174
Sara S Parker http://orcid.org/0000-0003-3670-6147
Samantha M Hill http://orcid.org/0000-0002-6454-7430
Robert A Grant http://orcid.org/0000-0002-5072-2867
Ghassan Mouneimne http://orcid.org/0000-0001-8103-4701
Amy E Keating http://orcid.org/0000-0003-4074-8980

### Decision letter and Author response

Decision letter https://doi.org/10.7554/eLife.70601.sa1
Author response https://doi.org/10.7554/eLife.70601.sa2

## Additional files

### Supplementary files

• Supplementary file 1. Protein sequences for A distributed residue network permits conformational binding specificity in a conserved family of actin remodelers.

• Supplementary file 2. Refinement table for ENAH-PCARE crystal structure.

• Transparent reporting form

### Data availability

Diffraction data have been deposited in the PDB with the accession code 7LXF. Source Data files have been provided for Figures 1, 2, 4, 5, and Table 1. These data files contain the numerical data used to generate the figures.

The following dataset was generated:

| Author(s) | Year | Dataset title | Dataset URL | Database and Identifier |
|---|---|---|---|---|
| Hwang T, Grant RA, Keating AE | 2021 | ENAH EVH1 domain bound to peptide from protein PCARE | https://www.wwpdb.org/pdb?id=pdb_00007lxf | RCSB Protein Data Bank, 7LXF |

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
