## [Editor Report]

This manuscript describes interesting follow-up studies on one peptide hit (from a protein called PCARE) coming out of a proteome-wide screen for peptides that can bind to the EVH1 domain of ENAH, one of the three highly homologous Ena/VASP actin regulators. Surprisingly PCARE binds to ENAH selectively over the other two members of Ena/VASP family, EVL and VASP. The authors provide a nice explanation for how this selectivity is achieved and develop a peptide PCARE-Dual that specifically binds ENAH more tightly, setting out the stage for developing potent and selective inhibitors for ENAH.

---

## [Decision Letter]

**Decision letter after peer review:**

Thank you for submitting your article "A distributed residue network permits conformational binding specificity in a conserved family of actin remodelers" for consideration by *eLife*. Your article has been reviewed by 3 peer reviewers, including Hening Lin as Reviewing Editor and Reviewer #1, and the evaluation has been overseen by Volker Dötsch as the Senior Editor. The following individual involved in review of your submission has agreed to reveal their identity: Linda Nicholson (Reviewer #2).

Essential revisions:

1) The Y62C mutant needs to be further characterized to rule out dimerization and to delineate the effect of V65P mutation.

2) For the structure, EVAH EVH1 is fused to the PCARE peptide. There is concern that the fusion may affect the binding interaction. Some experimental data is needed to address this concern.

3) Some experiments to show the physiological interaction between PCARE and EVAH is needed.

4) Revise the text/Figures according to reviewer suggestions.

These three points above are from the weaknesses identified by at least two reviewers and all three reviewers agreed during the discussion. The three reviewers felt that two co-submitted manuscripts could be merged into one much stronger manuscript as this manuscript provides some of the supporting data needed for the other manuscript.

*Reviewer #1 (Recommendations for the authors):*

I recommend the authors to revise the text to address the questions that I raised in the public review. Two of the comments need to be addressed with experimental data: the Y62C mutant binding data and the interaction between PCARE protein and ENAH protein.

*Reviewer #2 (Recommendations for the authors):*

1. Please add to the introduction a description of the Acevedo et al., 2017 work provides the basis for design of the dual-motif ligands. Titled "A Noncanonical Binding Site in the EVH1 Domain of Vasodilator-Stimulated Phosphoprotein Regulates Its Interactions with the Proline Rich Region of Zyxin", this paper identified and characterized the novel secondary binding site on the opposite face of the VASP-EVH1 domain that was exploited in design of the dual-motif ligands investigated in the current work.

2. The possible regulation of the interaction of ENAH with their dual-motif ligands by phosphorylation of ENAH should be addressed. An additional key finding of the Acevedo 2017 paper was that binding to the secondary site is inhibited by mutation of VASP EVH1 residue Y39 to E, providing a potential explanation for regulation of the VASP-Zyxin interaction by Abl-catalyzed phosphorylation of Y39. ENAH is also a reported target of Abl, and the VASP-Y39-equivalent residue is conserved in ENAH and EVL. The bivalent advantage the authors have designed in their dual-motif ligands could potentially be regulated by phosphorylation of this Y, with implications for their proposed use of a dual-motif ligand for "dissecting specific Ena/VASP functions in processes including cancer cell invasion" and "as a promising lead for developing therapies to treat ENAH-dependent diseases." It would provide essential context if the authors addressed this very important aspect.

3. It would strengthen the structural conclusions if it could be demonstrated that untethered PCARE peptide binds in the same orientation as observed in their crystal structure of the fusion protein. The main concern is that tethering the peptide to the C-terminus could constrain the binding options of the peptide (i.e., force a binding mode that differs from the biological interaction). For the VASP/Zyxin interaction, the application of NMR enabled the untethered and tethered binding interactions to be matched using chemical shift perturbations. Ideally, EVAH-EVH1 complexed with a tight-binding PCARE peptide (perhaps PCARE-Dual) could be crystallized and its structure determined. Otherwise, the possibility that fusion of the binding partners might influence the binding mode should be addressed.

4. Figure 2B: The quantification of colocalization in focal adhesions can be more rigorously done using ImageJ along with a plugin for colocalization. The cross-section plots "along a line through focal adhesions" show individual examples but do not provide the necessary statistics on the degree of colocalization observed for each of the three paralogs.

5. Figure 2F: does the 4th row in each of the mRUBY2-PCARE B and Mito-mRUBY2-PCARE B columns show visualization of mRUBY2? This should be labeled on the vertical edge, as the other 3 rows are. It would be helpful to keep the same vertical order of protein being visualized; currently the 4-row column is ordered Pax/ENAH/PCARE and the enlarged boxes are Pax/PCARE/ENAH.

6. It would be very helpful to add some discussion of the biological function of PCARE and its connection to actin dynamics.

*Reviewer #3 (Recommendations for the authors):*

The impact of this work could be enhanced if key elements from both manuscript were combined into a single, more comprehensive paper. This manuscript is relatively short but closely related to the first thematically in that it describes MassTitr-identified interaction features.

The authors should discuss the potential for the Y62C dimers to form, and rule out that they exist in their preparations.

---

## [Author Response]

Essential revisions:1) The Y62C mutant needs to be further characterized to rule out dimerization and to delineate the effect of V65P mutation.

We have made and tested EVL Y62C, for comparison to EVL Y62C V65P. Our biolayer interferometry experiments were carried out in a buffer that contained 1 mM DTT to prevent the formation of disulfide-mediated oligomers. We now also demonstrate that EVL Y62C runs at the same volume as EVL V65P in size exclusion chromatography, ruling out the possibility that this mutation leads to formation of a dimer or other higher-order state; see Supplementary File 1.

To test whether the higher affinity of EVL Y62C V65P for PCARE could be attributed to the Y62C mutation alone, we measured binding for EVL Y62C, and found K_D_ = 19 ± 2.8 μM; now included in Table 1. This value is almost unchanged from binding of wt EVL (K_D_ = 22 ± 1.5 μM), which demonstrates that this mutation is not sufficient to confer high-affinity of binding to PCARE on EVL. We describe this result and its implications on p. 5.

2) For the structure, EVAH EVH1 is fused to the PCARE peptide. There is concern that the fusion may affect the binding interaction. Some experimental data is needed to address this concern.

Covalently linking a motif-containing peptide to a domain that it binds to solve a structure is a standard approach, particularly in the short linear motif field. We added references on p. 4 to provide background about this point. But we shared the reviewers’ concern about the fusion potentially affecting the binding mode and took care to design a linker that would not constrain the binding. Specifically, our crystallization construct consisted of the ENAH EVH1 domain followed by a 6-residue Gly-Ser linker, and then a 36-amino acid segment from PCARE (PCARE^813-848^). In truncation experiments, we determined that the N-terminal 13 residues of the PCARE peptide (residues 813-825) are dispensable for the high-affinity binding and we now emphasize this point on p. 4-5. In addition, in our crystal structure, there is no continuous density for the Gly-Ser linker or for PCARE residues 813 – 825, consistent with the linker being disordered and not engaged with ENAH. In contrast, we see clear electron density for PCARE^828-848^, which coincides almost exactly with the regions that we identified as necessary for high affinity binding in solution. Thus, our biochemical experiments support what we see in the crystal structure.

We were further reassured by mutational experiments in EVL EVH1 domain that showed: (1) mutations in regions of the EVL EVH1 domain corresponding to those that contact the C-terminal tail of PCARE in the ENAH-PCARE structure affected PCARE binding but not ActA binding (i.e. EVL V65P/Y62C in Table 1), strongly suggesting that this region is involved in binding the C-terminal tail of PCARE, and (2) we successfully redesigned the EVL EVH1 domain to confer high-affinity binding of PCARE by making mutations entirely guided by the ENAH-PCARE structure.

However, because these mutations were made in EVL, not ENAH, in response to the reviews, we have made and tested a mutation to a residue of ENAH that contacts the C-terminal part of PCARE in the structure and is remote from the poly-proline region (which binds as expected in the canonical groove): mutation P65D. This point mutation leads to a negligible, 1.40-fold change in the binding affinity of ActA (a canonical FP4 binder), indicating that the structure of the domain remains intact, but it weakens binding of PCARE by 167-fold. The dissociation constant for the mutant, which we determined to be 32 ± 3 μM, is similar to that for other FP4-containing peptides that lack affinity enhancing elements. This is what we would predict, based on our structure, if PCARE lost the extra, non-canonical, affinity-enhancing interactions with ENAH that we observe. We chose the P65D mutation to be highly diagnostic of this binding mode, as shown in Figure 3. This additional experiment directly tests and supports the geometry that we observe in the high-resolution crystal structure. We have described the new mutant on p. 5 and added it to Table 1.

3) Some experiments to show the physiological interaction between PCARE and EVAH is needed.

An interaction between PCARE and ENAH has been reported in two separate affinity purification-mass spectrometry experiments, and the two proteins have also been shown to co-localize at ciliary expansions, providing excellent support for the physiological relevance of the biochemical interaction that we observe in this work. We put more emphasis on this synergistic work from cell-based studies on p. 2 and on p. 8.

4) Revise the text/Figures according to reviewer suggestions.

We describe changes that we have made to reviewer suggestions below.

*Reviewer #1 (Recommendations for the authors):*
I recommend the authors to revise the text to address the questions that I raised in the public review section. Two of the comments need to be addressed with experimental data: the Y62C mutant binding data and the interaction between PCARE protein and ENAH protein.

We have addressed these points as described above.

Reviewer #2 (Recommendations for the authors):1. Please add to the introduction a description of the Acevedo et al., 2017 work provides the basis for design of the dual-motif ligands. Titled "A Noncanonical Binding Site in the EVH1 Domain of Vasodilator-Stimulated Phosphoprotein Regulates Its Interactions with the Proline Rich Region of Zyxin", this paper identified and characterized the novel secondary binding site on the opposite face of the VASP-EVH1 domain that was exploited in design of the dual-motif ligands investigated in the current work.

This paper was very important for our work and we now highlight it in the Introduction, as well as in other parts of the paper, to make sure that the important context of our study is clear to readers.

2. The possible regulation of the interaction of ENAH with their dual-motif ligands by phosphorylation of ENAH should be addressed. An additional key finding of the Acevedo 2017 paper was that binding to the secondary site is inhibited by mutation of VASP EVH1 residue Y39 to E, providing a potential explanation for regulation of the VASP-Zyxin interaction by Abl-catalyzed phosphorylation of Y39. ENAH is also a reported target of Abl, and the VASP-Y39-equivalent residue is conserved in ENAH and EVL. The bivalent advantage the authors have designed in their dual-motif ligands could potentially be regulated by phosphorylation of this Y, with implications for their proposed use of a dual-motif ligand for "dissecting specific Ena/VASP functions in processes including cancer cell invasion" and "as a promising lead for developing therapies to treat ENAH-dependent diseases." It would provide essential context if the authors addressed this very important aspect.

We tested the affinity of PCARE-dual for ENAH Y38E and found that, as anticipated by the reviewer (and by us), this reduces affinity, consistent with disruption of the back-side site by introduction of a negative charge (Figure 5 —figure supplement 1). However, it is interesting that Tani et al., report that the site on ENAH that is phosphorylated by Abl (in the presence of Abi1) is not the Y39-equivalent residue Y38 but in fact Y296. We were not able to find any direct evidence of the phosphorylation of ENAH Y38. This raises the intriguing (but undemonstrated) possibility that different phosphorylation specificity, combined with dual-site inhibitors, might further enhance paralog specificity of molecules like PCARE-dual. We added the new data for ENAH Y38E and a discussion of this point on p. 6 and p. 9.

3. It would strengthen the structural conclusions if it could be demonstrated that untethered PCARE peptide binds in the same orientation as observed in their crystal structure of the fusion protein. The main concern is that tethering the peptide to the C-terminus could constrain the binding options of the peptide (i.e., force a binding mode that differs from the biological interaction). For the VASP/Zyxin interaction, the application of NMR enabled the untethered and tethered binding interactions to be matched using chemical shift perturbations. Ideally, EVAH-EVH1 complexed with a tight-binding PCARE peptide (perhaps PCARE-Dual) could be crystallized and its structure determined. Otherwise, the possibility that fusion of the binding partners might influence the binding mode should be addressed.

We presented the evidence that we have for the observed orientation in the section addressing the “essential revisions,” above. We would be excited to get a structure of one of the dual-motif peptides bound to ENAH, but we don’t have any promising leads for this currently.

4. Figure 2B: The quantification of colocalization in focal adhesions can be more rigorously done using ImageJ along with a plugin for colocalization. The cross-section plots "along a line through focal adhesions" show individual examples but do not provide the necessary statistics on the degree of colocalization observed for each of the three paralogs.

We have re-done this analysis and updated Figure 2B. See discussion on p. 3 of our revised co-localization analysis.

5. Figure 2F: does the 4th row in each of the mRUBY2-PCARE B and Mito-mRUBY2-PCARE B columns show visualization of mRUBY2? This should be labeled on the vertical edge, as the other 3 rows are. It would be helpful to keep the same vertical order of protein being visualized; currently the 4-row column is ordered Pax/ENAH/PCARE and the enlarged boxes are Pax/PCARE/ENAH.

We thank the reviewer for this helpful comment. We have updated Figure 2F based on these suggestions.

6. It would be very helpful to add some discussion of the biological function of PCARE and its connection to actin dynamics.

We agree with the reviewer and have added additional information on the known function of PCARE to the Introduction on p. 2.

Reviewer #3 (Recommendations for the authors):The impact of this work could be enhanced if key elements from both manuscript were combined into a single, more comprehensive paper. This manuscript is relatively short but closely related to the first thematically in that it describes MassTitr-identified interaction features.

We considered many ways of organizing our findings for clearest communication to researchers in relevant fields, including a single, large paper. We concluded (based on feedback from others, as well as our own impressions) that reporting the broader findings of the screen in a short report and then diving deeper into the interesting mechanism of PCARE specificity in an article would ensure that the detailed biochemistry and design work was not lost or made harder to find when presented as part of a broad proteomic survey. We are happy with how the two papers came out and hope the editor and reviewers will permit us this discretion in choosing how to present the work.

The authors should discuss the potential for the Y62C dimers to form, and rule out that they exist in their preparations.

Please see above.